# A Ratiometric Fluorescent Nano-Probe for Rapid and Specific Detection of Tetracycline Residues Based on a Dye-Doped Functionalized Nanoscaled Metal–Organic Framework

**DOI:** 10.3390/nano9070976

**Published:** 2019-07-04

**Authors:** Lei Jia, Shengli Guo, Jun Xu, Xiangzhen Chen, Tinghui Zhu, Tongqian Zhao

**Affiliations:** 1College of Chemistry and Chemical Engineering, Henan Polytechnic University, Jiaozuo 454000, China; 2Institute of Resources & Environment, Henan Polytechnic University, Jiaozuo 454000, China

**Keywords:** tetracycline, multi-color fluorescence, visual detection, lanthanide probe, metal–organic framework

## Abstract

Tetracycline (TC) residues are harmful to the environment and human body, so it is necessary to develop a highly sensitive probe for rapid detection of tetracycline residues. In the present paper, a novel dye-doped porous metal–organic framework (UiO-66)-based multi-color fluorescent nano-probe was designed for sensitive ratiometric detection of tetracycline (TC). In this probe, dye-molecules doped UiO-66 was used as a fluorescent internal standard, and the externally grafted lanthanide Eu^3+^ complex was used as response signals. The fluorescence of the Eu^3+^ complex was selectively enhanced with increasing concentrations of TC, which was accompanied by a visual blue-to-red color switch. The nano-probe had a linear response between 0.1 and 6 μM with a lowest detection limit of 17.9 nM, which was much lower than the maximum residue limits set by the United States Food and Drug Administration (676 nM) and the European Union (225 nM). The applicability of this method in the analysis of actual samples was evaluated by the determination of TC in honey and milk samples, indicating satisfactory recovery and good reproducibility. In addition, a cost-effective paper-based probe for rapid and visual detection of TC was developed by fixing the nano-probe on filter papers. With the help of a smartphone camera to capture the fluorescence color, and chromaticity analysis software, the calculation and analysis of red (R) and blue (B) values can be realized, which has the potential for real-time visual detection of TC.

## 1. Introduction

Since its inception in 1948, tetracycline (TC) has been widely used as a spectrum bacteriostatic agent for the treatment of human and animal infections [1]. However, the abuse of tetracycline in animal husbandry and aquaculture leads to excessive levels of residual tetracycline in food and the environment [2]. These residues may have adverse effects on the human body, including damage to the liver, kidneys and muscle tissue [3], and may also cause bacterial resistance to antibiotics [4]. Therefore, the detection of tetracycline in food and the environment is important to reduce human uptake of tetracycline.

In recent years, methods for detecting TC mainly contain high performance liquid chromatography (HPLC) [5], mass spectrometry (MS) [6], Raman spectroscopy [7], capillary electrophoresis [8] and microbial analysis [9]. However, most of these methods have some defects. As for HPLC, MS and Raman spectroscopy, they offer good separation and selectivity, but the requirement of professional skill and tedious operating procedures limits their application. Capillary electrophores have poor sensitivity and reproducibility. Microbial analysis is still unsatisfactory because of the potential cross-reactivity and high cost. Therefore, developing a low-cost, easy-to-operate detection method is an effort of current researchers.

Fluorescence-based detection methods, especially the lanthanide-based fluorescent probes, attract many researchers’ eyes because of their ease of operation, high sensitivity and short response time. It has been reported that TC containing β-diketone configuration can form stable luminescent complexes with europium [10,11]. However, the existence of coordination water molecules in the first coordination layer often leads to fluorescence quenching caused by the vibration of molecules [12]. In order to avoid the fluorescence quenching, some organic molecules can be used to replace coordinated water molecules and coordinate with Eu^3+^. The fluorescence intensity of nano-probes can be changed and the ultra-sensitive and specific detection of TC can be realized [13].

Researchers have developed some lanthanide-based on–off sensors to measure target concentrations [14,15]. However, most of these sensors measure the concentration of the target only by the change of fluorescence intensity, so the influence caused by instrument or environmental factors is inevitable [16]. The ratiometric fluorescent analysis method can significantly reduce these disturbances through built-in self-calibration and accurate measurements [17,18,19]. Selection of a fluorescent internal standard with good stability and no change with the concentration of the analyte is very important.

The metal–organic frameworks (MOFs) are a class of three-dimensional infinite network structure materials, which can be bridged by a metal core and an organic ligand, and have a high specific surface area, a large pore volume and a high adjustability [20,21,22]. In recent decades, MOFs have been applied to separation [23], chemical sensors [24], gas storage [25], catalysis [26] and biosensors [27,28]. For example, UiO-66 is a MOF material with high adsorption and large specific surface area, and the preparation method is mature and simple, which can be used as a scaffold in ratio fluorescent probes.

In this study, we developed a novel dye-doped UiO-66-based multi-color fluorescent nano-probe for TC detection (Scheme 1). As we know, fluorescent brightener has high thermal stability and bright color, which is an ideal dye for reference fluorescence [29]. The narrow micropore distribution of MOFs makes the dye molecules evenly dispersed and is not easy to agglomerate. Therefore, UiO-66 doped with dye molecules was used as a fluorescent internal standard in this research. Next, a multi-purpose silica shell was assembled for encapsulation of the internal standard. The silica shell not only effectively protected the dye molecules and reduced losses [30], but also provided the attachment of amino groups, which facilitated the grafting of europium complexes. Based on the above advantages, the fluorescent dye molecules were encapsulated into UiO-66 and covalently grafted with the europium complex, which can realize the multi-color fluorescent sensing of TC. With the addition of TC, TC continuously transferred energy to Eu^3+^ and the dominated ^5^D_0_ emission of Eu^3+^ increased, and the fluorescence color changed from blue to red, which facilitated detection and observation.

## 2. Materials and Methods

### 2.1. Materials and Reagents

All 1,4-benzenedicarboxylic acid, Fluorescent Brightener KS-N and Zirconium Chloride were purchased from Shanghai Titan Scientific Co., Ltd., China. *N*,*N*-dimethylformamide (DMF), citric acid, ammonia and silicic acid tetraethyl ester were purchased from Hedong District of Tianjin Hongyan Reagent Factory, Tianjin, China. (3-aminopropyl)triethoxysilane (APTES) was purchased from Shanghai Macklin Biochemical Co., Ltd., Shanghai, China. Ethanol was purchased from Shanxi TongJie Chemical reagent Co., Ltd., Xi’an, China. Polyvinylpyrrolidone (PVP), *N*-Hydroxysuccinimide (NHS), Dicyclohexylcarbodiimide (DCC), 3-(3-dimethylaminopropyl)-1-ethylcarbodiimide hydrochloride) (EDC) and Eu(NO_3_)_3_·6H_2_O were purchased from Shanghai Adamas Reagent Co., Ltd., Shanghai, China.

### 2.2. Preparation of UiO-66

UiO-66 was synthesized according to the reported method [31]. A total of 53 mg of ZrCl_4_ and 34 mg of terephthalic acid were dissolved in 25 mL DMF, and then the mixture was transferred to a Teflon reactor and heated at 120 °C for 24 h. After cooling to room temperature, the mixture was centrifuged to obtain precipitate, followed by washing several times with water, and dried under vacuum at 60 °C for 24 h to obtain a white powder.

### 2.3. Preparation of Dye (Fluorescent Brightener KS-N)@UiO-66-@SiO_2_-NH_2_

A total of 100 mg of UiO-66 and 0.1 mg of dye were added into a 20 mL ethanol solution. The mixture was stirred for 3 h and followed by adding 5 mg of polyelectrolyte PVP and 10 mL of distilled water, which is helpful to block the entrances of nanochannels of the NMOF and thus can prevent the leakage of the dyes from the nanochannels [32]. The above mixture solution was then stirred for 1 h and the obtained white solid was collected by centrifugation. The obtained white solid was then dispersed into 15 mL of ethanol, and 1 mL of a mixed solution containing APTES, TEOS and ammonia water (volume ratio was 10:1:20) was dropwise added in order. The mixture was stirred at room temperature for 8 h, and the resulting white precipitate can be obtained after centrifugation, washed three times with water and dried at 60 °C.

### 2.4. Preparation of Dye@UiO-66-@SiO_2_-NH_2_-Cit-Eu

To activate the carboxyl group of Cit, 10 mg of NHS, 10 mg of EDC, and 60 mg of Cit were dissolved in 20 mL of distilled water and stirred for 4 h. Dye@UiO-66-@SiO_2_-NH_2_ (50 mg) was then added and then stirred for 24 h at room temperature. The obtained precipitates were dispersed in ethanol for 30 min at 25 °C, followed by the addition of 12 mg Eu(NO_3_)_3_·6H_2_O. Then the mixture was heated in an oil bath at 60 °C for 5 h. The final products were obtained by centrifugation and washed several times with distilled water and ethanol, and dried at 60 °C for 12 h.

### 2.5. Determination of Standard TC Concentration in Honey and Milk Samples

As reported in the literature [33], purified milk was obtained from local markets and the processing method was as follows: 300 g/L trichloroacetic acid was added at 1% (*v*/*v*) in milk, sonicated for 20 min, and centrifuged to remove proteins and lipids. Then the supernatant was filtered with a 0.22 μm membrane, and a series of standard TC concentration samples were prepared using the filtered solution. Honey sold in the market was diluted 20 times, filtered with a 0.22 μm filter to prepare a series of standard concentrations of TC honey solution. The nano-probe was dispersed into Tris-HCl buffer to obtain a suspension, exposed to a tetracycline standard concentration aqueous solution, and the fluorescence spectrum was recorded. As for the actual sample detection, the same operation was performed.

### 2.6. Preparation of Test Paper Modified with Dye@UiO-66@SiO_2_-Cit-Eu for Visual Detection of TCs

Firstly, the common filter paper was cut into round strips (7 mm in diameter) and added to the dispersion of Dye@UiO-66@SiO_2_-Cit-Eu nano-probe (2 mg/mL), then incubated for 1 h and dried in air at 30 °C. After drying, the Dye@UiO-66@SiO_2_-Cit-Eu immobilized filter paper was used as the test paper for detection. Different concentrations of TC samples were dropped onto the prepared test paper and the fluorescent color changes of the paper under 365 nm UV light can be digitized and put out by using a developed color-scanning application.

### 2.7. Characterization

Transmission electron microscopy (TEM), high resolution transmission electron microscopy (HRTEM), and the energy dispersive spectra (EDS) were determined using a Tecnai-G2-F30 (FEI, Eindhoven, The Netherlands) at acceleration voltages of 200 kV. X-ray diffraction (XRD) measurements were carried out on an X’pert PRO X-ray power diffractometer (PANalytical Co.X’pert PRO, Almelo, The Netherlands) using Cu Ka radiation of 1.5406 A (40 kV, 30 mA). The UV measurement was finished on a Shimadzu UV-240 spectrophotometer (Shimadzu Corporation, Kyotp, Japan).

## 3. Results and Discussion

### 3.1. Structural and Morphology Characterization

The microstructures of pure UiO-66 and Dye@UiO-66@SiO_2_-Cit-Eu were characterized by transmission electron microscopy and the results are shown in Figure 1. Pure UiO-66 particles displayed cubic morphologies with particle sizes between 50 and 80 nm (Figure 1a), which was consistent with related reports [10,31]. The resulted Dye@UiO-66@SiO_2_-Cit-Eu had a core-shell structure (Figure 1b), with the particle diameter between 100 and 150 nm, and the silica shell was used to encapsulate the adsorbed fluorescent dye molecules. This structure can also effectively reduce dye loss and facilitate grafting with europium. The chemical composition of Dye@UiO-66@SiO_2_-Cit-Eu was investigated by an energy dispersive spectrometer (EDS). Clearly, from Figure 1c, Zr, Si, O and Eu elements were distributed on the nano-probe, demonstrating that the europium ions were successfully grafted onto the surface of UiO-66 (Figure 1c); the energy and the related counts for each element are also listed in Appendix A.

The XRD patterns of UiO-66, Dye@UiO-66@SiO_2_ and Dye@UiO-66@SiO_2_-Cit-Eu nanocomposites are shown in Figure 2a. It was obvious that the diffraction peaks of pure UiO-66 were consistent with previous reports (111, 002, 224 and 046 diffractions) [31,34]. After the dye molecules were adsorbed and encapsulated with silica, the position of the diffraction peak of UiO-66 did not change, indicating that the crystal structure of the nanomaterials were well maintained. After coating with amorphous silica, there was a cladding peak at 2θ = 20°–30° [35]. Except for the decrease in diffraction intensity, there was no significant difference between the diffraction peak of the UiO-66, Dye@UiO-66@SiO_2_ and Dye@UiO-66@SiO_2_-Cit-Eu, thereby indicating that the crystal structure of UiO-66 is well preserved and the interactions occur on the exterior surface, but not in the interlayer.

To further demonstrate the successful preparation of the probe, the samples were characterized by FTIR and the results are shown in Figure 2b. The two strong absorption peaks of UiO-66 at 1580 cm^−1^ and 1400 cm^−1^ were related to the in-phase and out-of-phase tensile modes of carboxylate groups [36]. The absorption peaks at 400–800 cm^−1^ were due to the mixing of the –OH and –CH bending modes with the Zr–O mode at lower frequencies [36]. The strong absorption peak of Dye@UiO-66@SiO_2_ at 1622 and 1050 cm^−1^ after encapsulation with silica shell can be ascribed to the Si–OH vibrations and the asymmetric stretching vibration of Si–O–Si [37], which may cover part of the fine structure of UiO-66. The bands at 1715 and 1652 cm^−1^ of Dye@UiO-66@SiO_2_-Cit-Eu can be attributed to the –COOH and –CO–NH vibrations. All the above results proved that the europium citrate complex (Cit-Eu) had been successfully grafted onto the surface of Dye@UiO-66@SiO_2_ [38].

### 3.2. Sensitivity and Selectivity of Dye@UiO-66@SiO_2_-Cit-Eu Nano-Probe for Detection of Tetracycline

A multi-color fluorescence system was designed to detect tetracycline in this research. As shown in Figure 3a, the Dye@UiO-66@SiO_2_-Cit-Eu nano-probe had a blue fluorescent emission from dye molecules. When residual tetracycline existed in the testing samples, the Dye@UiO-66@SiO_2_-Cit-Eu can capture tetracycline to form a Dye@UiO-66@SiO_2_-Cit-Eu-TC complex. After the coordination between tetracycline and Eu^3+^, the nano-probe emitted a characteristic red light at 617 nm through the antenna effect, while the blue fluorescence emission of the dye molecules encapsulated by UiO-66 (430 nm) remained unchanged and it can be used as a reference. By recording the change in the ratio of the red emission of the Eu–TC complex to the blue emission of the Dye@UiO-66@SiO_2_-Cit-Eu, the external environmental interference can be effectively reduced, and a highly sensitive detection of tetracycline can be achieved.

In order to explore the effect of pH on the sensitivity of the probe in the environment, the nano-probe was placed in a series of different pH buffer solution to investigate the change of fluorescence intensity in the presence of TC (Figure 3b). It can be seen that the fluorescence emission peak at 430 nm was almost unchanged, while the fluorescence emission peak at 617 nm changed most obviously with the change of pH. With the increase of pH value, the β-diketone configuration in TC loses protons, and the increase of deprotonated TC was beneficial to the formation of the Eu–TC complex. When the pH value was greater than 9.0, the fluorescence intensity of Eu–TC decreases, which may be due to the formation of europium hydroxide precipitation under alkaline conditions [12]. The fluorescence ratio of the red emission to the blue emission was plotted against the pH value, which indicated the highest sensitivity of the nano-probe at pH = 9.

To evaluate the sensitivity of the Dye@UiO-66@SiO_2_-Cit-Eu nano-probe for the detection of tetracycline, we added the probe to different concentrations of TC in a tris-HCl buffer solution at pH = 9. As can be seen from Figure 4a, the emission peak at 430 nm wavelength was used as the contrast fluorescence, and the fluorescence intensity was substantially constant. The emission peak at 617 nm was narrow and sharp, which was the peak with the fastest change in TC concentration and the fluorescence intensity increased with increasing TC concentration. In addition, the Dye@UiO-66@SiO_2_-Cit-Eu nano-probe was also a visual probe for detecting TC residues. When the concentration of TC was increased under the irradiation of UV light with a wavelength of 365 nm, multicolor fluorescence was observed by the naked eye, and the fluorescence changed from blue to red (Figure 4b), which had a higher resolution than that of “off–on” fluorescence. The color coordinated of the Dye@UiO-66@SiO_2_-Cit-Eu nano-probe for various concentrations of TC were calculated (Appendix A) and the chromaticity diagram (Figure 4c) also clearly showed a linear fluorescence variation of the Dye@UiO-66@SiO_2_-Cit-Eu probe from blue to red. The ratio of peaks at 617 nm and 430 nm can be calculated to reflect the change in TC concentration. As the TC concentration increased from 100 nM to 6 μM, the intensity ratio I_617_/I_430_ increased linearly with a correlation coefficient of 0.998 (Figure 4d), and the detection limit calculated based on a signal-to-noise ratio of 3 was 17.9 nM, which was lower than many of the previously reported methods, such as electrochemical analysis based on a molecularly imprinted carbon nanotube composites (49.5 nM) [39], electrochemiluminescence based on silica nanoparticles (160 nM) [40] and fluorescence based on adenosine monophosphate (60 nM) [41]. Meanwhile, the interference of intensity fluctuations caused by environmental or instrumental factors can be eliminated by ratio fluorescence. The high sensitivity, convenient use and fast response speed (1 min) gave Dye@UiO-66@SiO_2_-Cit-Eu a clear advantage in terms of sensing performance.

In order to apply this probe to the actual sample detection, the selectivity of the nano-probe during the TC detection was then investigated. In general, a milk sample mostly contains amino acids, common metal ions and other substances. Therefore, under the optimized conditions, these coexisting substances, including L-arginine (Arg), L-aspartic acid (Asp), L-cysteine (Cys), L-glutamic acid (Glu), L-histidine (His), L-lysine (Lys), Ca^2+^, Cu^2+^, K^+^, Mg^2+^ and Hg^2+^ were individually added to the reaction solutions, and the intensity ratio I_617_/I_430_ was calculated. It is shown in Figure 4e that these substances had a slight effect on fluorescence, which indicated that the method here had a high selectivity for TC detection.

### 3.3. Application in the Actual Samples

The above experiments indicated that the Dye@UiO-66@SiO_2_-Cit-Eu nano-probe still maintained good selectivity and sensitivity in the presence of a variety of common impurities in milk. Therefore, the real-time detection of TC in milk samples will have certain research significance for food safety detection. Herein, honey and milk were chosen as the actual samples. A certain amount of pure TC was added to the above samples, and a series of actual samples with different TC concentration gradients were obtained. The actual TC concentrations were determined with the Dye@UiO-66@SiO_2_-Cit-Eu nano-probe. As listed in Table 1, the recoveries were in the range of 97.05–106.01%, and the relative standard deviations (RSD, n = 3) were in the range of 0.59–4.91%. The results showed that the method we used had relatively stable precision and good reproducibility.

### 3.4. Visual Detection of TC Based on Test Paper

In order to make the detection of TC more simple and rapid, a cost-effective paper-based probe for rapid and visual detection of TC was developed by fixing the nano-probe on filter paper. Under the UV lamp (Ex = 365 nm), it can be seen that the fluorescence color of the paper changed from blue to red depending on the concentration of TC (Figure 5). With the help of a smartphone camera to capture the fluorescence color and chromaticity analysis software, the calculation and analysis of red (R) and blue (B) values can be realized. When the ratio of R to B in the image was Y and the concentration of TC was X, a fitting curve can be obtained when the concentration of TC was 0–5 μM, and the linear equation was y = 0.2775x + 0.3076. The minimum detection concentration of the paper-based vision probe was about 0.1 μM, which was much lower than the maximum residue limits of 0.676 μM and 0.225 μM for TC in milk set by the United States Food and Drug Administration (FDA) and the European Union (EU) [42]. The above results showed that the nano-probe can realize the real-time visual detection of TC in pure products and actual samples and this study had certain potential application value for the quantitative and semi-quantitative detection of TC in other actual samples.

## 4. Conclusions

In summary, a visual dye-doped multi-color fluorescent nano-probe for highly sensitive and selective detection of TC by using metal–organic frameworks (UiO-66) as a scaffold was developed. The fluorescence color of the nano-probe changed from blue to red depending on the concentration of TC. This proposed nano-probe was also applied to the determination of TC in honey and milk samples with satisfactory results. In addition, a Dye@UiO-66@SiO_2_-Cit-Eu coating test paper combined with a smartphone chromaticity analysis application is expected to achieve rapid, real-time and on-site analysis of TC.

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
