# Peer review of "A Ratiometric Fluorescent Nano-Probe for Rapid and Specific Detection of Tetracycline Residues Based on a Dye-Doped Functionalized Nanoscaled Metal–Organic Framework"

_nanomaterials, 2019, doi:10.3390/nano9070976_

Reviewer 1 Report

This study describes the development of a multi-color fluorescent nano-probe for fast and specific detection of TC. The fluorescence of the europium complex selectively increases with increasing concentrations of TC and is associated by a visual blue to red color shift. The study is of high importance and relevant for the journal. Overall, the data is clear and the conclusions are well supported by the data. 

However, there are frequent spelling and grammatical errors and the manuscript must be revised to improve the language. 

Author Response

Answer: Thanks for your positive evaluation and your evaluation will inspire us in our subsequent research. We have checked the manuscript sentences by sentences and modified the manucript word by word.

Reviewer 2 Report

1) The title could be modified for making it clearer to highlight the core novelties of the study e.g. by replacing “Multi-Color Fluorescent” with “Ratiometric fluorescence” or by adding “as internal standard” at the end of the title.

2) Lines 22-23: What is the level of TC in real samples? Can the obtained LR and LOD support it?

3) Line 44-45: The sentence “However, most of these methods require expensive instruments, the detection process is cumbersome, and the sensitivity is low” must be rewritten to make the advantages and disadvantages clearer. For example: MS analysis provides sensitive detection. However, it is an expensive instrument which needs experienced personal and….

4) Lines 50-54: Please rewrite the sentences with more explanation to make them clearer.

5) Line 62: What is the meaning of “infinite” here?

6) Line 82-84: Please rewrite the sentence “the energy transfer of fluorescent probe from TC to Eu3+ was gradually enhanced, and the red light emitted by Eu3+  gradually dominated”.

7) Line 104: “followed by adding 5 mg of PVP and 10 mL of distilled water.” Why PVP and water are added?

8)  Line 111: Pleases add: “To activate the carboxyl group of Cit,  10 mg of NHS and 10 mg of EDC were added to …”.

9) Line 119: Before spiking, was the real sample analyzed for TC?  Added 1% (v/v) seems to be too much, isn’t it?

10) Figure1c:  Could you please provide a small table besides the figure contacting energy and the related counts for each element?

11) Lines 156-157: “the interaction between the 156 Dye@UiO-66@SiO2 and the Cit-Eu occurred on the surface of UiO-66.”. How is it concluded? More explanation is needed?

12) Lines 188-189: “It can be seen that the fluorescence emission 188 peak at 617 nm changed most obviously with the change of pH,”. Pleased add explanation based on the structure.

13) Figure 4e and lines 227-229: What is the concentration of added extra substances? Please make the figure 4.e clearer.

14) Line 224: “by fixing the nano-probe on filter paper.”. How??? What is the detailed information for the prepared paper-based sensor?

Author Response

1) The title could be modified for making it clearer to highlight the core novelties of the study e.g. by replacing “Multi-Color Fluorescent” with “Ratiometric fluorescence” or by adding “as internal standard” at the end of the title.

Answer: Thanks for your kindly advice and we have modified the title as “A Ratiometric Fluorescent Nano-Probe for Rapid and Specific Detection of Tetracycline Residues Based on a Dye-Doped Functionalized Nanoscaled Metal-Organic Framework”.

2) Lines 22-23: What is the level of TC in real samples? Can the obtained LR and LOD support it?

Answer: Thanks for your brilliant suggestion and it is really a good question. The nano-probe had a linear response between 0.1-6 μM with a lowest detection limit of 17.9 nM, which was much lower than the maximum residue limits of 0.676 μM and 0.225 μM for TC in milk set by the United States Food and Drug Administration (FDA) and the European Union (EU) (     Tan, H.L.; Ma, C.J.; Song, Y.H.; Xu, F.G.; Chen, S.H.; Wang, L. Determination of tetracycline in milk by using nucleotide/lanthanide coordination polymer-based ternary complex. Biosens. Bioelectron. 2013, 50, 447-452).

3) Line 44-45: The sentence “However, most of these methods require expensive instruments, the detection process is cumbersome, and the sensitivity is low” must be rewritten to make the advantages and disadvantages clearer. For example: MS analysis provides sensitive detection. However, it is an expensive instrument which needs experienced personal and….

Answer: Thanks for your kindly advice and we have modified the sentences: As for HPLC, MS and Raman spectroscopy, they offer good separation and selectivity, but the requirement of professional skill and tedious operating procedure limits their application. Capillary electrophores has poor sensitivity and reproducibility. Microbial analysis is still unsatisfactory because of the potential cross-reactivity and high cost.

4) Lines 50-54: Please rewrite the sentences with more explanation to make them clearer.

Answer: Thanks for your kindly advice and we have rewrote the sentences: However, the existence of coordination water molecules in the first coordination layer often leads to fluorescence quenching. In order to avoid the fluorescence quenching caused by the vibration of coordinated water molecules, some organic molecules can be used to replace coordinated water molecules and coordinate with Eu3+. The fluorescence intensity of nano-probes can be changed and the ultra-sensitive and specific detection of TC can be realized.

5) Line 62: What is the meaning of “infinite” here?

Answer: Thanks for your good suggestion. There is an infinite extension of three-dimensional network structure in the microstructure of MOF materials.

6) Line 82-84: Please rewrite the sentence “the energy transfer of fluorescent probe from TC to Eu3+ was gradually enhanced, and the red light emitted by Eu3+  gradually dominated”.

Answer: Thanks for your brilliant suggestion. With the addition of TC, TC continuously transfers energy to Eu3+, and the dominated 5D0 emission of Eu3+ increases.

7) Line 104: “followed by adding 5 mg of PVP and 10 mL of distilled water.” Why PVP and water are added?

Answer: Thanks for your brilliant suggestion. It has been reported that coating the dye-loaded porous material initial with polyelectrolyte (e.g., poly(vinylpyrrolidone) (PVP) and subsequently with a silica shell by means of sol-gel processes can block the entrances of nanochannels of the porous material and thus can prevent the leakage of the dyes from the nanochannels (A. Guerrero-Martinez, S. Fibikar, I. Pastoriza-Santos, L. M. Liz-Marzάn, L. De Cola, Angew. Chem.Int. Ed. 2009, 48, 1266).

8)  Line 111: Pleases add: “To activate the carboxyl group of Cit, 10 mg of NHS and 10 mg of EDC were added to …”.

Answer: Thanks for your kindly advice and we have rewrote the sentences.

9) Line 119: Before spiking, was the real sample analyzed for TC?  Added 1% (v/v) seems to be too much, isn’t it?

Answer: Thanks for your brilliant suggestion. Prior to the experiment, we did not find any acycline in milk samples because milk samples were purchased directly from the market and tested by the health sector. While the description of the amount of 1% (v/v) may be a little less detailed. As reported in the literature (J. Hazard. Mater. 2015, 299, 132-140), purified milk was obtained and 300 g/L trichloroacetic acid was added at 1% (v/v) in milk, so the addition of trichloroacetic acid was still a little.

10) Figure1c:  Could you please provide a small table besides the figure contacting energy and the related counts for each element?

Answer: Thanks for your brilliant suggestion. Because EDX element analysis can only analyze the composition of the material qualitatively, from figure 1c we can see that Zr, Si, O and Eu elements were distributed on the nano-probe, demonstrating that the europium ions were successfully grafted onto the surface of UiO-66. We haso had listed the energy and the related counts for each element in Table S2.

11) Lines 156-157: “the interaction between the Dye@UiO-66@SiO2 and the Cit-Eu occurred on the surface of UiO-66.”. How is it concluded? More explanation is needed?

Answer: Thanks for your brilliant suggestion. PXRD studies showed that Dye@UiO-66@SiO2 and Dye@UiO-66@SiO2-Cit-Eu exhibited the same pattern as UiO-66, proving the presence of the NMOF core in both PVP- and silica-coated UiO-66. Except for the decrease in diffraction intensity, there was no significant difference between the diffraction peak of the Dye@UiO-66@SiO2 and Dye@UiO-66@SiO2-Cit-Eu, thereby indicating that the crystal structure of UiO-66 is well preserved and the interactions occur on the exterior surface, but not in the interlayer.

12) Lines 188-189: “It can be seen that the fluorescence emission peak at 617 nm changed most obviously with the change of pH,”. Pleased add explanation based on the structure.

Answer: Thanks for your brilliant suggestion and we have added the explanation listed below: With the increase of pH value, the β-diketone configuration in TC loses protons, and the increase of deprotonated TC is beneficial to the formation of Eu-TC complex. When the pH value is greater than 9.0, the fluorescence intensity of Eu-TC decreases, which may be due to the formation of europium hydroxide precipitation under alkaline conditions.

13) Figure 4e and lines 227-229: What is the concentration of added extra substances? Please make the figure 4.e clearer.

Answer: Thanks for your kindly advice and we have added the concentration of added extra substances.

14) Line 224: “by fixing the nano-probe on filter paper.”. How??? What is the detailed information for the prepared paper-based sensor?

Answer: Thanks for your brilliant suggestion and this is really a good question. Firstly, the common filter paper was cut into round strips (7 mm in diameter) and added to the dispersion of Dye@UiO-66@SiO2-Cit-Eu nano-probe (2 mg/mL), then incubated for 1 h and dried in air at 30. After drying, the Dye@UiO-66@SiO2-Cit-Eu immobilized filter paper was used as test paper for detection. Different concentrations of TC samples were dropped onto the prepared test paper and the fluorescent color changes of the paper under 365 nm UV light can be digitized and output by using a developed color-scanning application.

Thanks a lot again for your good suggestions and carefully review.

Reviewer 3 Report

In this MS, Jia et al. report preparation of novel MOF-based material for detection of tetracycline traces in various products. Overall, design of all experiments is correct, and all conclusions are well supported by the data. Characterization of the material is done in compliance with the modern standards. Also, the style of the text is good; only minor spellchecking may be needed.

In my opinion, this work is well suitable for Nanomaterials journal.

Author Response

In this MS, Jia et al. report preparation of novel MOF-based material for detection of tetracycline traces in various products. Overall, design of all experiments is correct, and all conclusions are well supported by the data. Characterization of the material is done in compliance with the modern standards. Also, the style of the text is good; only minor spellchecking may be needed. 

In my opinion, this work is well suitable for Nanomaterials journal.

Answer: Thanks for your positive evaluation and your evaluation will inspire us in our subsequent research. We have checked the manuscript sentences by sentences and modified the manucript word by word.